# Experimental Study on the Features of Infrasonic Waves of Sandstone under Shear Load

**Chen Qiao** [1,2,3], **Fenglin Xu** [4], **Pengcheng Su** [5], **Yang Liu** [2], **Yifang Zhang** [2], **Honglin Zhu** [6], **Haitao Huang** [7], **Man Huang** [3,*] , **Jilong Chen** [2] **and Dunlong Liu** [8]

1 Key Laboratory of Mine Geological Hazard Mechanism and Prevention and Control, Xi'an 710054, China; chenqiao@cigit.ac.cn
2 Chongqing Institute of Green Intelligent Technology, Chinese Academy of Sciences, Chongqing 400714, China; liuyang125152@163.com (Y.L.); zhangyifang@cigit.ac.cn (Y.Z.); chenjilong@cigit.ac.cn (J.C.)
3 Key Laboratory of Rock Mechanics and Geohazards of Zhejiang Province, Shaoxing College of Arts and Sciences, Shaoxing 312000, China
4 Chongqing Environmental Protection Engineering Technology Center for Shale Gas Development, Chongqing 408000, China; xufenglin8@163.com
5 Institute of Mountain Hazards and Environment, Chinese Academy of Sciences, Chengdu 610041, China; supengcheng@imde.ac.cn
6 Graduate School, Chongqing Technology and Business University, Chongqing 400067, China; zhuhonglin@ctbu.edu.cn
7 Emergency Management Bureau of Emeishan City, Emeishan City 614200, China; haitao19851224@163.com
8 College of Software Engineering, Chengdu University of Information and Technology, Chengdu 610225, China; ldl@cuit.edu.cn
* Correspondence: hmcadx@126.com

**Abstract:** The shear failure of rock is a major cause of rock slope instability and consequent landslides. To determine the forming mechanism of infrasonic waves during the loss of stability of sandstone slopes, experiments were carried out using a shear loading device and an infrasonic monitoring device. In the experiments, infrasonic wave events were identified, and the characteristic parameters of infrasonic waves were extracted to analyze the features of the infrasonic wave response during the shear failure of sandstone. The study results show that: (1) the whole process of shear failure was associated with infrasound events. A normalized energy cumulative coefficient of over 0.6 and a normalized infrasound rate of over 0.89 are the key time nodes for alarming landslide; (2) with an increase in sample size, the shear resistance of the sample increases logarithmically, the total energy of infrasound events increases exponentially, and the average dominant frequency of infrasound events decreases linearly; and (3) with an increase in axial pressure, the shear of the rock increases almost linearly, the number of infrasound events increases linearly, and the average dominant frequency of infrasound events increases exponentially. The research results provide important guidance for the dynamic monitoring and evaluation of the stability of sandstone slopes and can provide a theoretical reference for landslide alarming of sandstone slopes using infrasonic waves.

**Keywords:** sandstone; shear load; infrasonic wave; automatic identification; geologic hazard

## 1. Introduction

During deformation and break under stress, rocks send out sounds [1–3]. The infrasonic wave associated with the sound, with low frequency and low attenuation, can bypass barriers [4], so it can be used to monitor and detect ruptures inside rocks. The formation of landslides due to the destruction of rocky slopes is a mechanic process; therefore, infrasonic waves, as a means of monitoring rock landslides, has drawn increasing attention from researchers in China and abroad.

Through a uniaxial compression experiment, Chai Shan, et al. [5] found that the rupture of granite generated infrasonic signals which could be used for monitoring rock

stability. Zhu Xing et al. [6,7] monitored the infrasonic wave signals during the whole loading process of six typical kinds of rock samples in real time and processed and analyzed the infrasonic wave signals to figure out the range of characteristic frequencies of infrasonic waves generated during rock failure. Xu Hong et al. [8] analyzed the energy features of infrasonic anomalies generated during failure of rock, and the study results provide an important basis for early warning of rock failure. To monitor and warn of coal dynamic hazards by infrasonic waves, Jia B. [9–13] and Wei J. P. [14] carried out a significant body of research. Through uniaxial loading experiments on coal samples, they collected infrasonic signals, analyzed the relative energy and frequency of the signals, and found that the relative energy and frequency of the infrasonic signals have obvious stage features. This laid the foundation for the stability monitoring of coal mine shafts based on infrasound. As the study went on, the team led by Zhao Kuai of Jiangxi University of Science and Technology [15–18] examined the effects of loading path, cementation and rock grain size on the infrasonic signal features, analyzed the features of infrasonic signals before and after peak uniaxial stress, and described the destruction process of granite by using the variation features of fractal dimension of infrasonic waves' energy rate. In addition to an experimental study on infrasonic waves in lab, Moran [19], Zhu X [20], Zimmer [21], and Moore [22] monitored the infrasonic arrays of rockfalls of Mount St. Helens in Washington, the side wall of Dagurangbao landslide, Yosemite Valley and Bingham Canyon, and analyzed the basic features of the infrasound of rockfall and the falling trajectory of rock. Allstadt [23] investigated the effects of different sizes, spaces, and types of volcanoes on the infrasonic signal and envisioned that the infrasonic waves would be a new means for monitoring the surface movement of volcanoes. The above study results have provided a theoretical reference for the infrasonic wave monitoring of rocky landslides.

Rocky landslides come mainly from shear destruction mechanically. At present, without systematic studies on the features of infrasonic waves during rock shear destruction, the features of infrasonic waves during rocky hill destruction remain unclear, and the forming mechanism of infrasonic waves in rocky landslides is not understood well, hindering the identification and prediction of rocky landslides. These questions are bottleneck issues that need to be solved urgently to realize infrasound monitoring and early warning of rocky landslide. Therefore, the features of infrasonic waves generated during sandstone shear destruction were investigated in this work, which is of great significance for the monitoring and early warning of rocky landslides.

## 2. Experiment to Acquire Infrasonic Signals during Sandstone Shear Destruction

### 2.1. Experimental System

To determine the features of infrasonic waves generated during sandstone shear destruction, a set of experimental apparatus testing the infrasonic waves generated by rock under shear loading was built (Figure 1), mainly including the rock shear loading system and infrasound acquisition system.

### 2.1.1. Rock Shear Loading System

The equipment used for rock shear loading was a YZ-50 direct shear apparatus with a maximum load of 1000 kN. The direct shear apparatus can complete rock shear loading under stable axial pressure, automatically collect data such as time, tangential force, and tangential displacement, and store data in the PC connected for later analysis.

① Infrasound signal acquisition system；②Rock shear loading system；
③Infrasound sensor；④Infrasound data collector；⑤Rock direct shear instrument

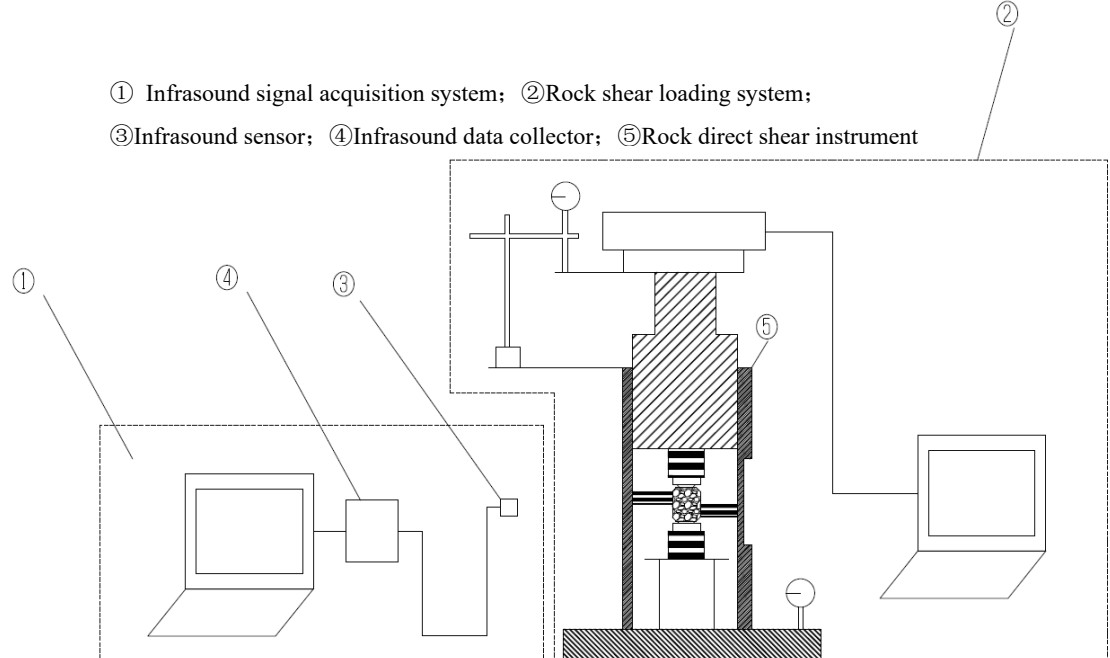

**Figure 1.** Experiment apparatus.

### 2.1.2. Infrasound Signal Acquisition System

The infrasound signal acquisition system includes an infrasound sensor and an infrasound data collector. The infrasound sensor is an IDS2016 infrasound sensor developed by the Institute of Acoustics, Chinese Academy of Sciences, with a sensibility more than 50 mV/Pa, coincidence error less than +/−1 dB, and a frequency band from 0.5 Hz to 300 Hz@3 dB, which covers the whole band range of infrasounds produced by the shear failure of rock. The infrasound data collector employs the nSYS2008 digital network transport protocol, which can pick up the signals received by the infrasound sensors synchronically at high speed through a local area network and transform the soundwaves into digital signals for later analysis. The sampling frequency of the experiments in this study was 100 Hz.

### 2.2. Rock Sample Preparation and Experiment Process

As a kind of sedimentary rock, sandstone plays an important role in many faults or weak layers in geotechnical engineering, especially as the stability of slip and weak zones in many slope projects is closely related to sandstone [24–26]. Sandstone is widely distributed in the area south of the Yangtze River. When constructing expressways, it appears in large quantities on the slopes in the area south of the Yangtze River. For example, the mileage of the Hunan Hengzao highway [27] through sandstone areas exceeds 100 km, and the maximum slope height is about 70 m. In the entire slope, landslides of different scales have developed, which have seriously affected the safety of the expressway's operation. Therefore, this article chooses sandstone as the experimental object. At present, the monitoring and early warning of sandstone slopes mainly use methods based on displacement, hydrological parameters, and mechanical parameters. These methods have a smaller monitoring range, while infrasound monitoring has a wider range [7]. The authors want to use this research to explore the possibility of infrasound methods for monitoring and early warning of sandstone landslides.

### 2.2.1. Rock Sample Preparation

(1) To determine the size effect of sandstone slopes, synchronous monitoring experiments of infrasounds generated during sandstone shear destruction were conducted on

sandstone samples of different sizes. A group of cubic samples with side lengths of 50, 100, 150, and 200 mm were prepared. Faces of the samples were planished. The samples were coded as a1-1, a1-2, a2-1, a2-2, a3-1, a3-2, a4-1, and a4-2 (Table 1).

**Table 1.** Basic parameters of sandstone samples and experiment design.

| No. | Size/mm$^3$ | Density/(g.cm$^{-3}$) | Saturation/% | Axial Pressure/Mpa |
|---|---|---|---|---|
| a1-1 | 50 × 50 × 50 | 2.49 | 0.00 | 0.00 |
| a1-2 | 50 ×50 × 50 | 2.34 | 0.00 | 0.00 |
| a2-1 | 100 × 100 × 100 | 2.45 | 0.00 | 0.00 |
| a2-2 | 100 × 100 × 100 | 2.50 | 0.00 | 0.00 |
| a3-1 | 150 × 150 × 150 | 2.41 | 0.00 | 0.00 |
| a3-2 | 150 × 150 × 150 | 2.43 | 0.00 | 0.00 |
| a4-1 | 200 ×200 × 200 | 2.44 | 0.00 | 0.00 |
| a4-2 | 200 × 200 × 200 | 2.44 | 0.00 | 0.00 |
| b1 | 50 × 50 × 50 | 2.49 | 0.00 | 0.00 |
| b2 | 50 × 50 × 50 | 2.55 | 0.83 | 0.00 |
| b3 | 50 × 50 × 50 | 2.44 | 1.67 | 0.00 |
| b4 | 50 × 50 × 50 | 2.25 | 2.50 | 0.00 |
| b5 | 50 × 50 × 50 | 2.52 | 3.33 | 0.00 |
| b11 | 50 × 50 × 50 | 2.37 | 0.00 | 0.00 |
| b21 | 50 × 50 × 50 | 2.39 | 0.83 | 0.00 |
| b31 | 50 × 50 × 50 | 2.42 | 1.67 | 0.00 |
| b41 | 50 × 50 × 50 | 2.42 | 2.50 | 0.00 |
| b51 | 50 × 50 × 50 | 2.45 | 3.33 | 0.00 |
| c1 | 50 × 50 × 50 | 2.49 | 0.00 | 5.00 |
| c2 | 50 × 50 × 50 | 2.47 | 0.00 | 10.00 |
| c3 | 50 × 50 × 50 | 2.45 | 0.00 | 15.00 |
| c4 | 50 × 50 × 50 | 2.43 | 0.00 | 20.00 |
| c5 | 50 × 50 × 50 | 2.38 | 0.00 | 25.00 |
| c11 | 50 × 50 × 50 | 2.52 | 0.00 | 5.00 |
| c21 | 50 × 50 × 50 | 2.44 | 0.00 | 10.00 |
| c31 | 50 × 50 × 50 | 2.32 | 0.00 | 15.00 |
| c41 | 50 × 50 × 50 | 2.40 | 0.00 | 20.00 |
| c51 | 50 × 50 × 50 | 2.45 | 0.00 | 25.00 |

(2) Rainfall is the most important coefficient that causes landslides. To determine the effect of rainfall on the sandstone slope, synchronous monitoring experiments of infrasounds generated during sandstone shear destruction were carried out on the rock sample with different water saturations. The water cut of rock sample was calibrated by the weighing method.

In this method, the first step is to put the sandstone samples of the same size into the oven for drying and dehydrate the core. During this period, the balance is continuously used to measure the change of rock sample quality until the rock sample quality no longer changes and the rock sample reaches the pure dry state. Then, the mass of the rock sample is recorded as M1. Second, after the rock sample cools, it is placed in a bucket to absorb water, and the water level exceeds the height of the rock sample. During this period, the balance is continuously used to measure the change of rock sample quality until the rock sample quality does not change and the rock sample reaches the state of pure water saturation. At this point, the mass of the rock sample is recorded as M2. The third step is to calculate the mass difference ΔM between the saturated and dry states of each rock sample and divide the difference into five different saturations: M1, M1 + 1/4ΔM, M1 + 2/4ΔM, M1 + 3/4ΔM, M2. The fourth step is to encapsulate pure dry rock samples and pure water-saturated rock samples with sealed bags. Other rock samples are placed in the oven to dry, and the mass changes of rock samples are continuously measured with a balance until the mass of rock samples reaches the weight of rock samples corresponding to the required saturation. These samples are then likewise encapsulated with sealed bags.

Sandstone samples with different water saturations were prepared by the water absorption and weighing method and coded as b1, b11, b2, b21, b3, b31, b4, b41, b5, and b51.

(3) In order to discuss the influence of different overburden pressures on sandstone slopes, infrasound synchronous monitoring experiments were carried out during the shear failure process under different axial pressures. A total of cubic sandstone samples 50 mm in side length were prepared and put into the oven to ensure the samples were dry. These samples were coded as c1, c11, c2, c21, c3, c31, c4, c41, c5, and c51.

Prepared sandstone samples are shown in Figure 2. Sandstone can be classified according to its permeability. The sandstone used in this study has a low permeability and belongs to the category of tight sandstone.

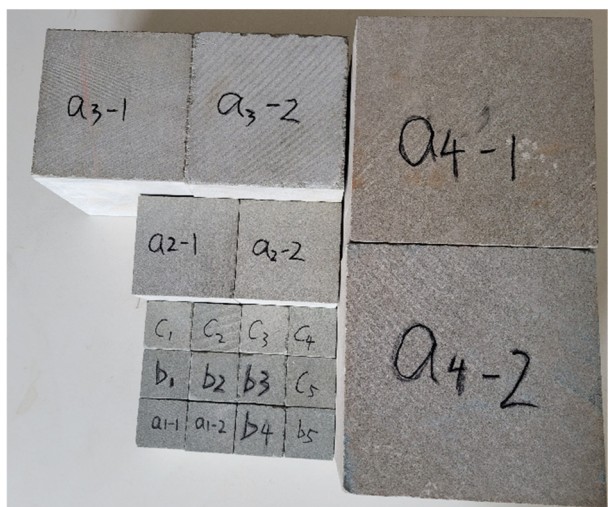
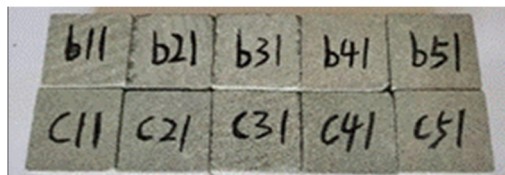

**Figure 2.** Sandstone samples.

### 2.2.2. Experiment Process

Different from the acoustic emission sensor, as the infrasound has low frequency, small energy attenuation, and long propagation distance, the infrasound sensor should not be in contact with the tested sample. Therefore, the sensor was fixed 200 mm away from the sample.

A set of straight cut boxes suitable for samples with side lengths of 50, 100, 150, and 200 mm were designed to apply to the YZ-50 direct shear apparatus (as shown in Figure 3). On this basis, we carried out shear breaking experiments on rock samples. The specific experimental steps are as follow. Firstly, the rock sample was put into the straight cut box; secondly, a set amount of normal stress was applied to the sample in the axial direction by the normal driving force arm; thirdly, the upper straight cut box was fixed by the fixed arm; finally, the driving arm exerted shear force on the bottom straight cut box until the rock sample was broken.

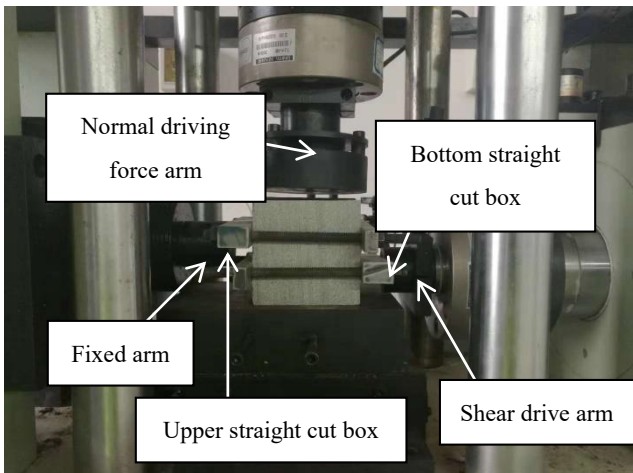

**Figure 3.** Shear loading on a rock sample.

### 3. Signal Processing and Analysis Method

*3.1. Analysis and Processing of Background Noise*

3.1.1. Noise Preprocessing

During the lab experiment, a lot of outside interference may appear, such as, for example, unstable air flow caused by opening or closing the door and exhaust fan operation, and ultra-low-frequency sounds produced by the water used in the experiment. To reduce these outside interferences caused by the controllable low-frequency sound sources mentioned above, the experiments were done in the wee hours and electrical devices not relevant to the experiments were all switched off.

3.1.2. Analysis of Background Noise

After noise preprocessing, the background noises in the experiments were mainly the sounds produced by mechanical vibration of the mechanical system during force loading. Therefore, before the experiment, the background noises produced by the mechanical system were collected by the infrasound sensor, and continuous wavelet transform and time-frequency analysis of the background noises were conducted (Figure 4). As shown in Figure 4, the background noise in the laboratory had relatively stable frequency, no outstanding frequency components, and low intensity ($\leq$0.022 dB).

3.1.3. Processing of Background Noises

Since this research mainly focused on the infrasounds in soundwave signals, the soundwaves need to be processed by a low-pass filter. In this study, the signals were processed with a Butterworth filter, in which the passband cutoff frequency was set at 20 Hz, the stopband cutoff frequency was set at 25 Hz, the maximum attenuation in the passband was 0.5 dB and the stopband attenuation was 40 dB. Meanwhile, in light of the low intensity of background noise, the intensity threshold of time frequency was set. Finally, the wavelet signals after threshold processing were put through inverse wavelet transform. As shown in Figure 5, the soundwave signals in the red dashed line box represent the infrasounds produced during shear loading. It is easier to acquire the frequency range of the infrasound events in the time-frequency spectrum.

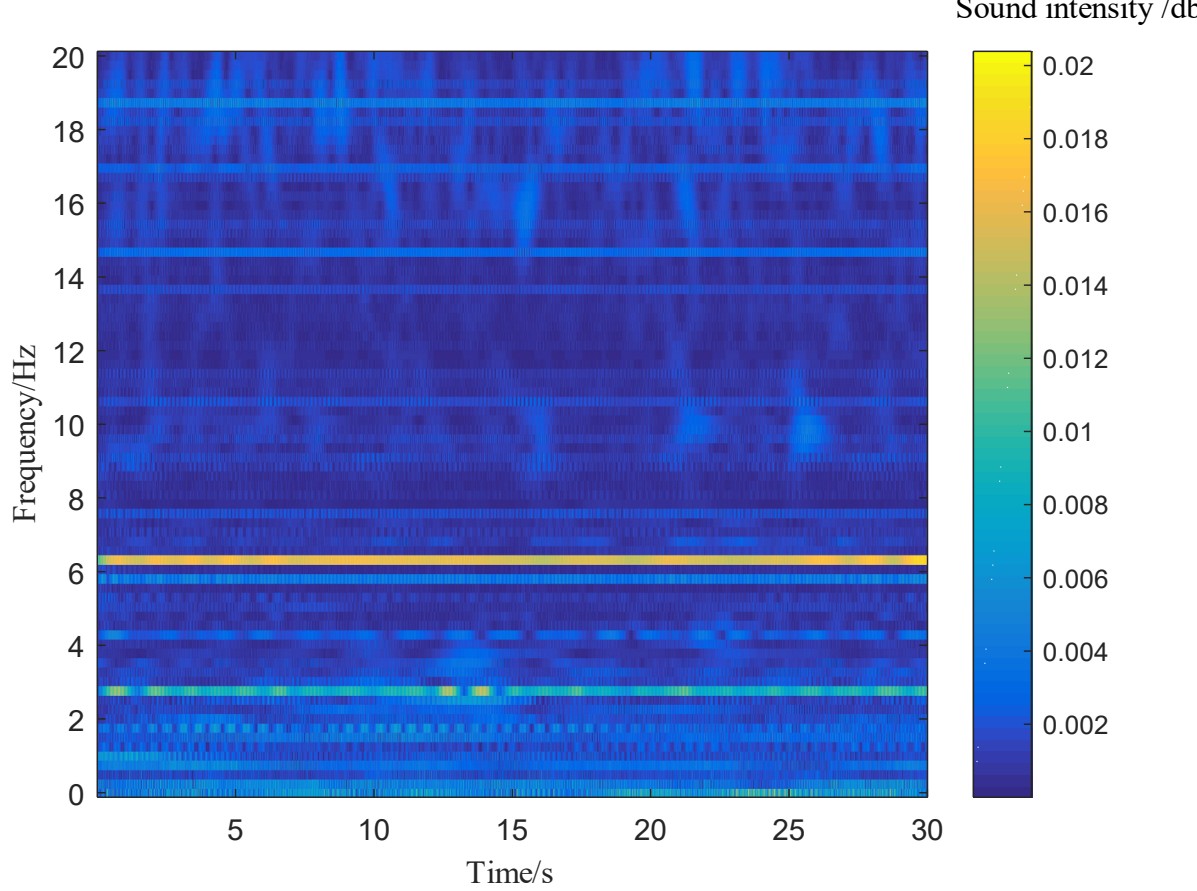

**Figure 4.** Time-frequency analysis of background noise.

### 3.2. Automatic Collection of Infrasound Events

Here, the concept of an infrasound event is introduced. In the whole time domain of a single experiment, effective infrasound signals did not exist all the time, but were intermittent in the time domain. Theoretically, an infrasound event can be defined as the process of an infrasound signal increasing from zero to a peak and then decreasing to zero in amplitude in the time domain. An infrasound event can be extremely complex in form. Infrasound events with big progressive increases in amplitude and significant energy concentration can be recognized by the naked eye, but it is difficult for humans to recognize the start and end of infrasound events with little increase in amplitude and low energy.

To collect infrasound events and divide the time domain according to infrasound events, we proposed an automatic time domain division method based on infrasound events with a computer program. In this method, first, whether a sampling point is in the time range of an infrasound event is judged based on its SNR. Then, the dense effective sampling points are merged into an infrasound event. The detailed steps are as follows [28]:

**Step 1.** If a sampling point is in the time range of an infrasound, the event is judged.

The energy formula of a discrete soundwave signal can be expressed as:

$$E = \frac{s}{\rho v} \sum_{i=1}^{I} x_i^2 \tag{1}$$

where $E$ represents the signal energy, $s$ represents the area passed by the soundwave (can be considered as a constant), $\rho$ represents the density of the medium through which the sound wave transmits, the air density herein, $v$ represents the sound speed, $I$ represents

the total number of sampling points, and *x(i)* represents the sound pressure value of the *i*th sampling point.

The energy of a single sampling point can be expressed as:

$$\Delta E = \frac{x_i^2 s}{\rho v} \tag{2}$$

According to the definition of power, in this study, the power of the *i*th sampling point is defined as:

$$P_i = \frac{x_i^2 s}{\rho v} \tag{3}$$

Then the criterion can be defined as:

$$\sum_{j=i-\lfloor \frac{L}{2} \rfloor}^{i+\lceil \frac{L}{2}-1 \rceil} \sqrt{P_j} > L\sqrt{P_{\max}} \tag{4}$$

where $P_j$ represents the power of the *j*th sampling point (the square of amplitude of the *j*th sampling point in the new signal), $P_{max}$ represents the maximum power of the first 30 s of the background noise (the square of the amplitude of the first 30 s of the new signal), and *L* represents the time window length to compute SNR (with totally *L* sampling points).

The left part of (6) represents the signal energy of time range *L* centered on the *i*th sampling point, while the right part represents the maximum energy of noise in time range *L*. This inequation represents the comparison of local signal energy with noise energy in the same time range. Every sampling point of the new signal is substituted into the inequation (6) to judge; if the result is real, then the point is marked as 1, and if not, the point is marked as 0. Thus, a sequence of 0 and 1 is generated and named the first marking sequence. The length of the sequence is equal to the original signal. At this time, the whole time domain is roughly divided based on SNR. The division still needs to be refined, since the "1" points in the first marking sequence are not concentrated enough. Signals which are notably close to each other need to be merged again.

**Step 2.** The first marking sequence is merged.

In this step, a row vector *r* whose value was 1 was used as the reference vector. Using the reference vector as a time window and panning it along the time domain of the first marking sequence, the part that has the same length with the time window can be computed as:

$$c_i = \frac{1}{J} \sum_{j=i-\lfloor \frac{J}{2} \rfloor}^{i+\lceil \frac{J}{2}-1 \rceil} s_j \cdot r_{j-i+\lfloor \frac{J}{2} \rfloor+1} \tag{5}$$

where $c_i$ represents the value that is used to judge if the sampling point (the *i*th point) needs to be remarked as 1 or 0, *J* represents the total number of sampling points of the sequence (which is also the window length or the length of *r*), $s_j$ represents the *j*th point of the first marking sequence, and $r_{j-i+\lfloor \frac{J}{2} \rfloor+1}$ represents the $j - i + \lfloor J/2 \rfloor + 1$th value of the reference vector *r*.

After $c_i$ is calculated from Equation (5), $c_i$ is then used to discern whether the sound pressure value of the sampling point needs to be reset. If $c_i > 0$, it indicates that the local vector in the first marking sequence corresponds to the reference vector, and the point can be marked as 1. If not, such that ($c_i < 0$), the point should be marked as 0. Finally, a new sequence of continuously distributed "0"s and "1"s based on the time domain is acquired and named the infrasound event marking sequence. In this sequence, a series of "1" indicates that, in this time range, an infrasound signal with significantly more energy than background noise exists, while a series of "0" indicates no effective infrasound signal in this time range.

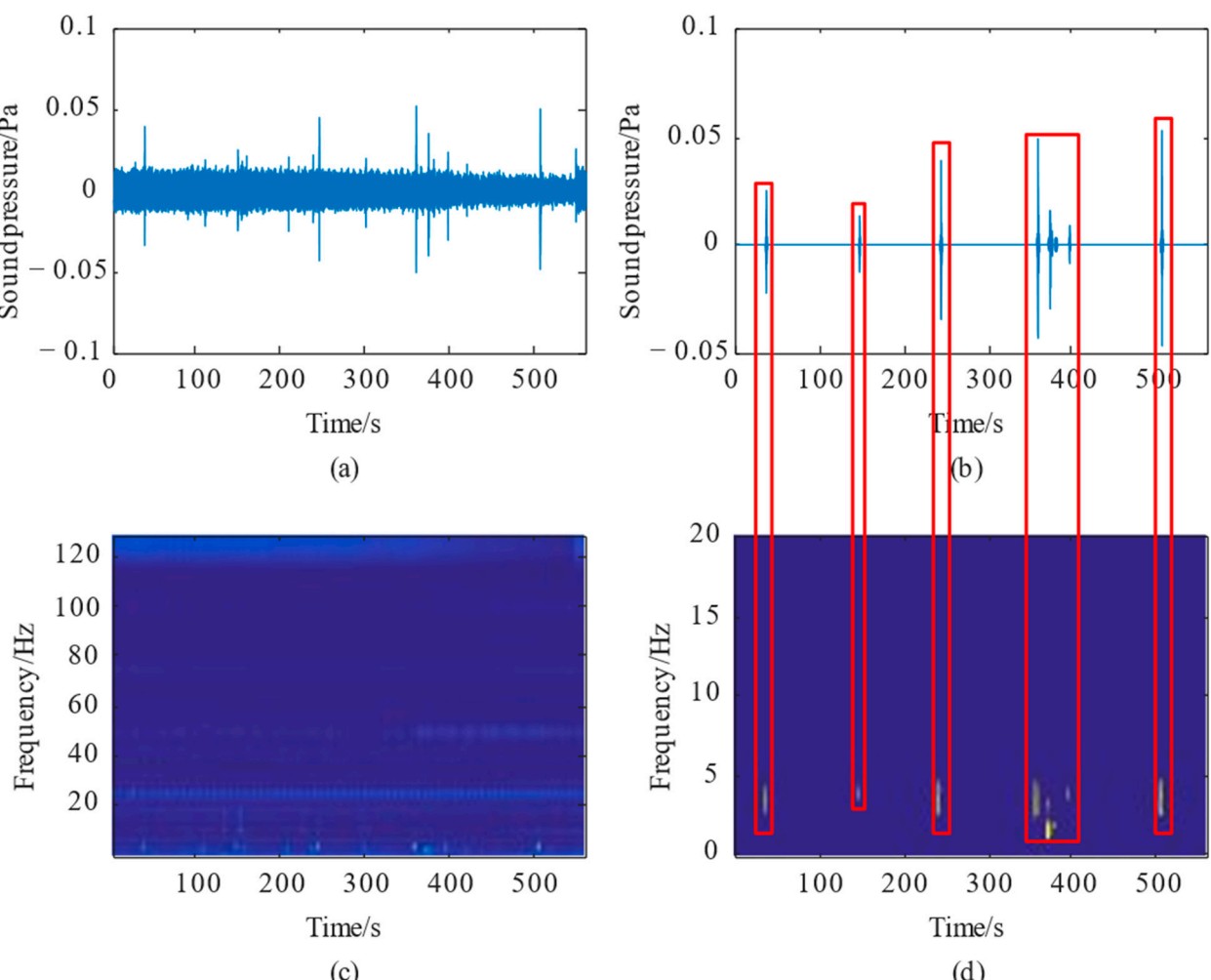

**Figure 5.** Signals before and after background noise processing. (**a**) Original signals, (**b**) signals after background noise processing, (**c**) time-frequency spectrum of the original signals, (**d**) time-frequency spectrum after background noise processing.

### 3.3. Definition of Infrasound Signal Energy

Amplitude indicates the vibration range of the signal, which can reflect how much energy the signal has. During rock shear failure, the energy of infrasound signals is mainly reflected by amplitude [29]. The amplitude can be expressed as:

$$E = \int |f(t)|^2 \, dt \tag{6}$$

where $E$ represents the total energy of the infrasound signal, and $f(t)$ represents the amplitude of the infrasound signal.

Based on the automatic collection of infrasound events, the total energy of a discrete signal can be represented as:

$$E = \sum_i^N |f(t_i)|^2 \tag{7}$$

where $f(t_i)$ represents the amplitude of the infrasound signal in the $i$th time range, and $N$ represents the number of infrasound events automatically recognized.

To determine the change in infrasound energy in the whole process of rock shear failure, this study defined the normalized energy accumulation rate as:

$$E_i = \sum_i^{N_i}|f(t_i)|^2 / \sum_i^{N}|f(t_i)|^2 \tag{8}$$

where $N_i$ represents the serial number of automatically recognized infrasound events according to time.

## 4. Analysis of Experiment Results

Three groups of typical experiment data were selected to separately plot the diagrams of the rock sample's normalization of the infrasound event over time, normalization of the energy accumulation rate over time and shear force over time (Figure 6). Since the straight shear load in the experiments was loaded with a constant change rate of displacement, the process of rock shear failure can be divided into four stages [30]: the compression stage, the elastic stage, the plastic stage, and the destruction stage. The last three stages correspond with the three key moments of the sandstone landslide [31]: forecast, early warning and warning (the turquoise vertical dashed line, purple vertical dashed line, and yellow vertical dashed line in Figure 6).

Overall, under straight shear loading, infrasound events always occur during sandstone destruction. In the experiment, the rock sample was first under normal stress, and its original opening structure plane or microfissures closed up gradually. When the stress was increased to shear stress, the rock sample was further compacted, and the grains inside the sample were squeezed and embedded into each other. Then, the shear stress increased slowly with the increase in strain, and the normalized energy accumulation coefficient also increased. When the shear stress increased further, the rock sample entered the elastic stage, and the microfissures in the sample underwent elastic deformation and propagated steadily. If the shear stress stopped at this point, the rock sample would stop breaking. The shear stress-strain curve in this stage is like a straight line after a sharp increase of the gradient. The normalized energy accumulation coefficient correspondingly rose sharply and then steadily. Therefore, the point that the normalized energy accumulation coefficient rises sharply can be taken as the start point of the forecast. With the increase in shear stress, the shear stress entered the yield point. The rock sample had qualitative changes, including microfissures, large deformation, cumulative damage, and the rapid increase of axial strain. When the sandstone is loaded in the yield stage, it simulates the imminent sliding stage of the sandstone landslide. The acoustic data and mechanical data of this experiment are jointly analyzed. The normalized energy accumulation coefficient at this stage is between 0.6 and 0.75, and the normalized infrasound changes between 0.89 and 0.93. Therefore, the values of these two acoustic parameters can be used to determine whether the sandstone is in the near-failure stage. At the same time, this feature can be used as a precursor feature of the impending sliding of a rock slope.

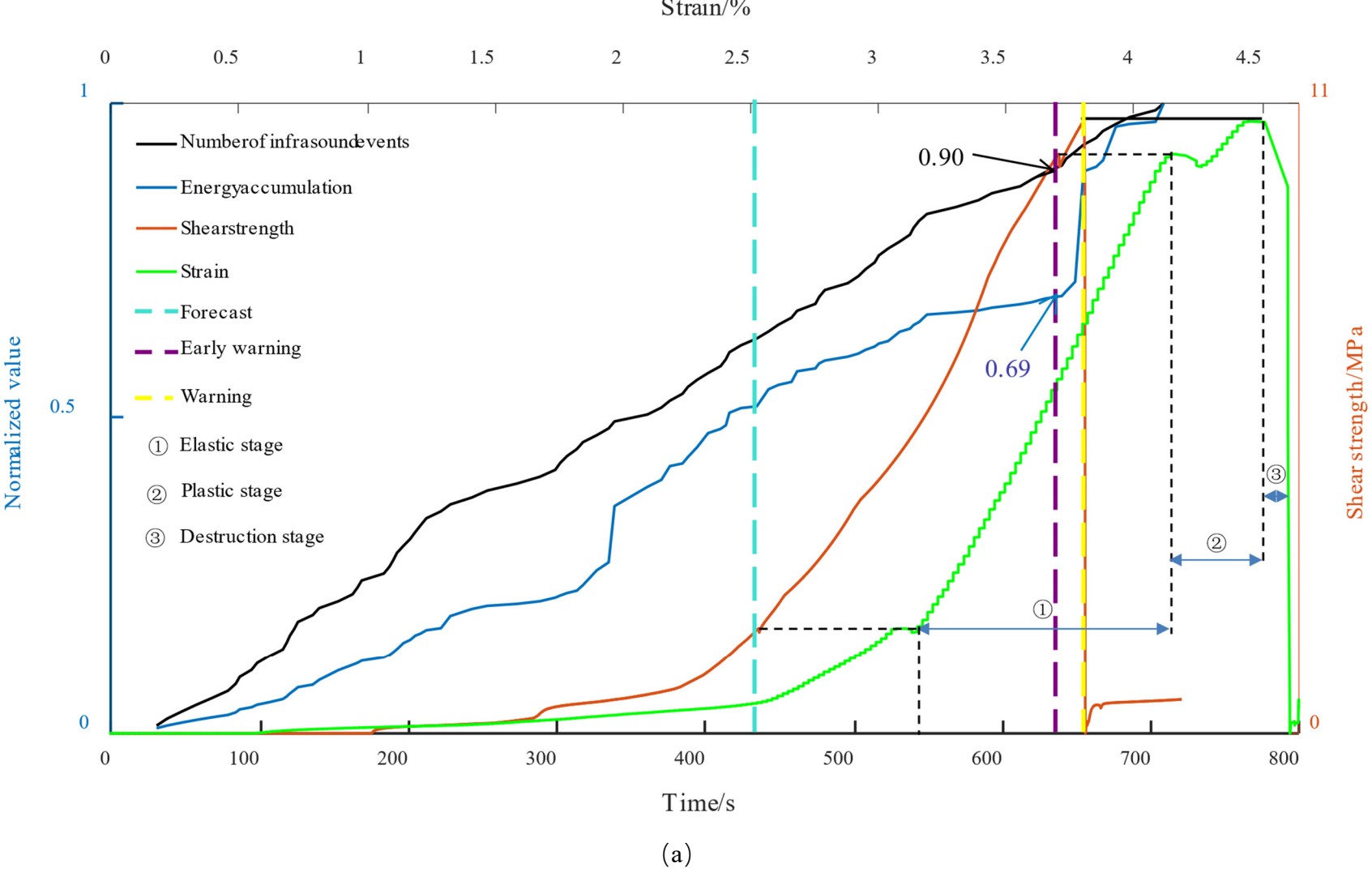

(a)

**Figure 6.** *Cont.*

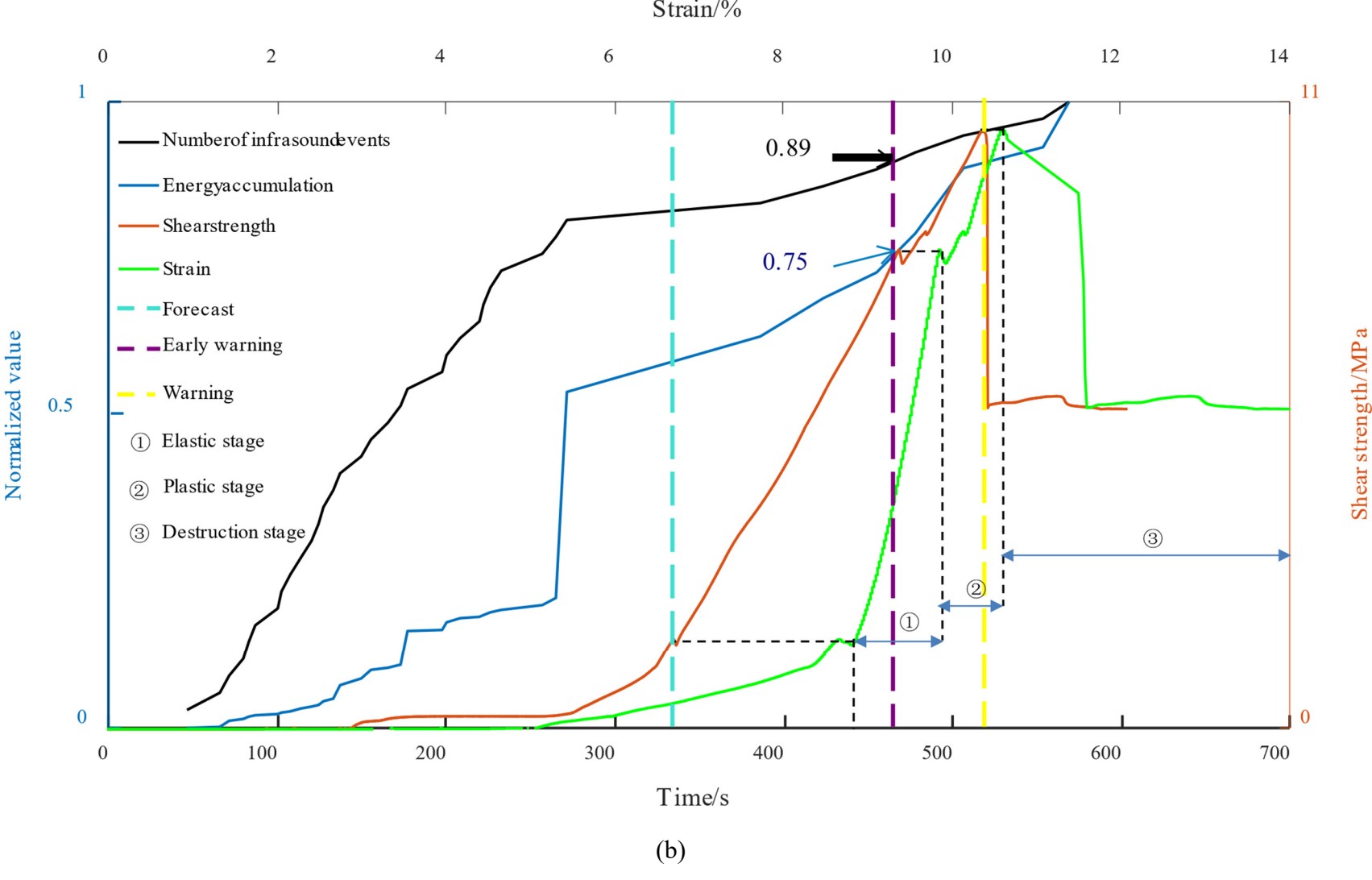

(b)

**Figure 6.** *Cont*.

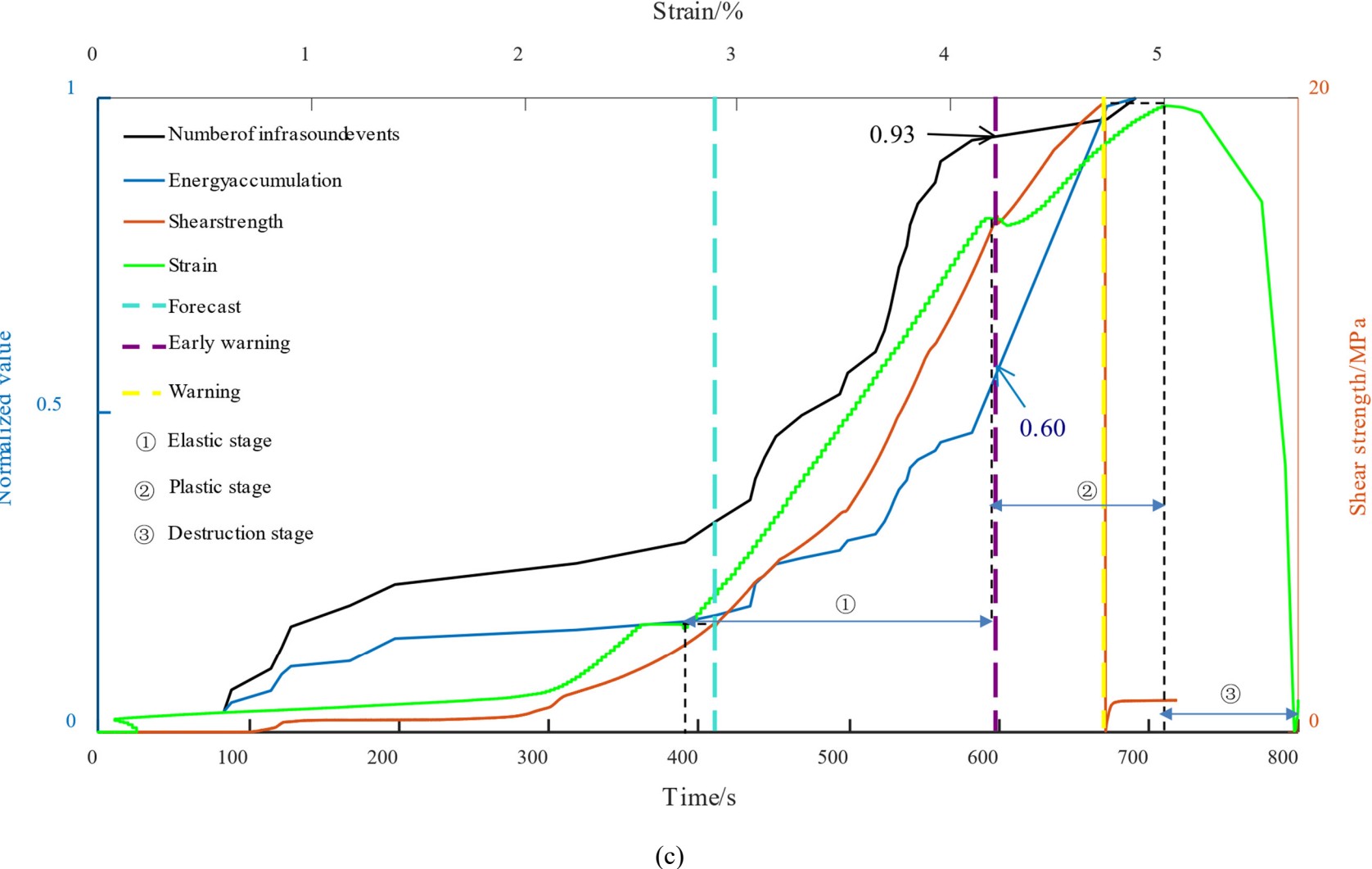

**Figure 6.** Relationships of normalized infrasound event rate, normalized energy accumulation rate, and shear force-time diagram of typical samples. (**a**) Sample a1-1; (**b**) sample b2; (**c**) sample c3.

*4.1. Impact of Sample Size on Infrasound Features in Shear Destruction*

From the perspective of mechanics, during shear destruction, with an increase in sample size, the shear resistance strength of the rock sample increases logarithmically (as shown in Figure 7). We found that the total infrasound energy of rock sample a4-1 is 6 times higher than that of rock sample a4-2. However, from the photos of the test results, the failure of the a4-1 rock sample produced fewer, longer cracks, and the sounds created by the failure of the rock sample are mainly concentrated in the low-frequency range. In contrast, the failure of sample a4-2 produced fractures larger in number but shorter in length, and the sounds generated by the rock sample failure are mainly concentrated in the high-frequency range. When damaged by the same load, the two rock samples produced similar total acoustic emission energy. Therefore, the acoustic emission energy generated by the failure of rock sample a4-1 in the infrasound frequency range must be higher than that of rock sample a4-2. Therefore, when the sandstone is damaged under the same shear load under the same size condition, the factor that has the greatest influence on the infrasound energy of the rock sample is the integrity of the sandstone damage.

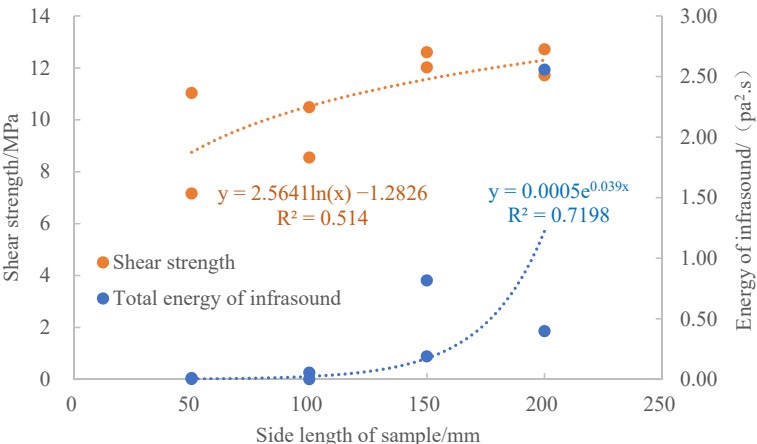

**Figure 7.** Relationship of shear resistance strength and infrasound energy amount.

From the photos of the experiment results (Figure 8), when the sample is small (as shown in Figure 8a,b), the failure surface is flat and low in roughness, the damage of the shear plane is mainly slip, and it appears as mostly surface wear. When the sample increases in size (as shown in Figure 8c,d), the damage surfaces become jagged, including two types: the serrated type and the flat type. Meanwhile, the sample have cracks which appear and the damaged surface increases in roughness. Destruction in this stage is mainly shear destruction.

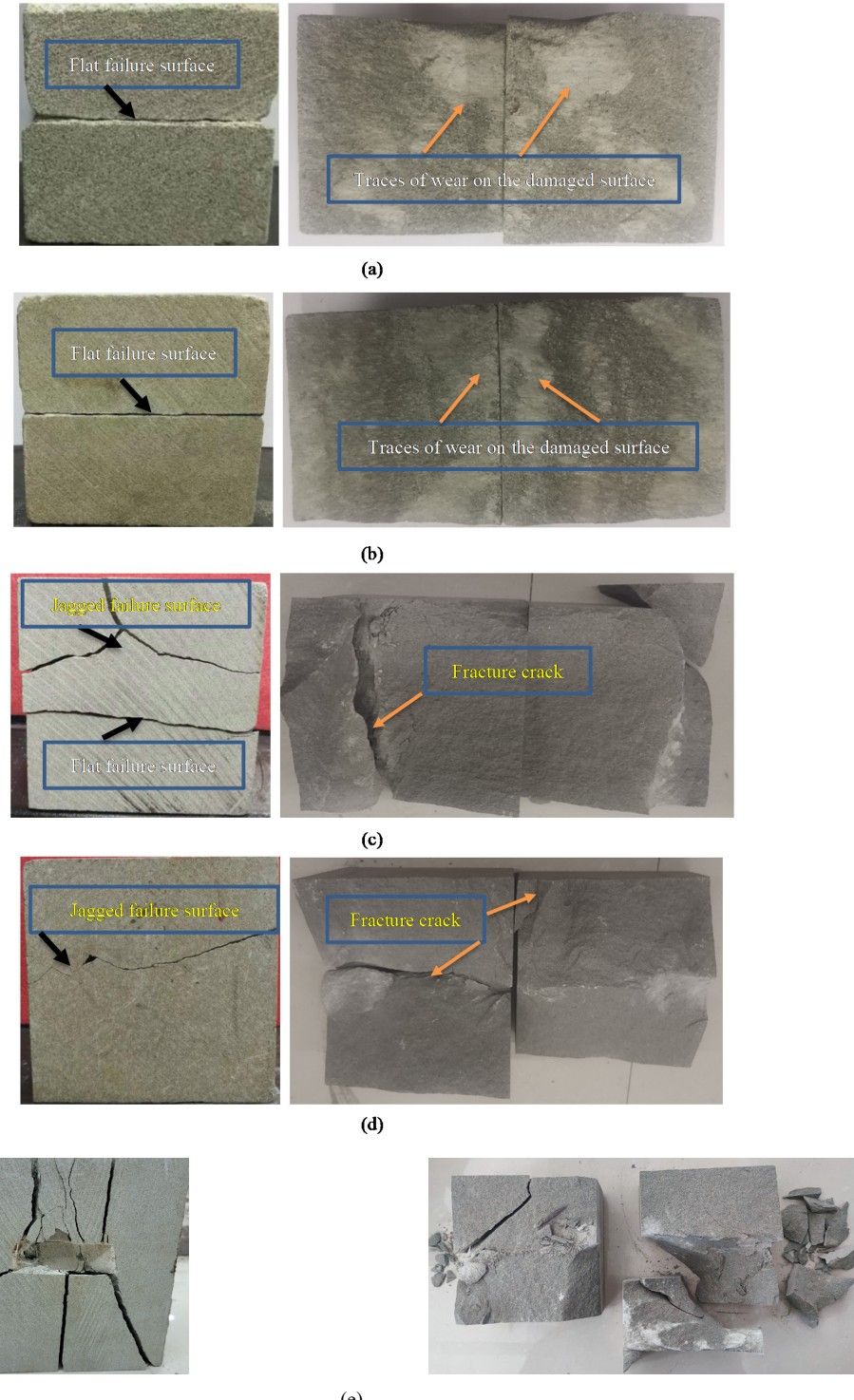

**Figure 8.** Straight shear experiment results of samples of different sizes. (**a**) Sample a1-1; (**b**) sample a2-1; (**c**) sample a3-1; (**d**) sample a4-1; (**e**) sample a4-2.

From the perspective of acoustics, with the increase in sample size, the total energy of infrasound events increased logarithmically (as shown in Figure 7). The number of infrasound events has little regularity in variation, but the average excellent frequency of infrasound events decreased linearly (as shown in Figure 9). During the shear loading of rocks, small cracks gradually joined together into big cracks, and the excellent frequency decreased with the increase of crack length [32]. Therefore, with the increase in sample

size, the cracks caused by loading increased in length, and the corresponding excellent frequency of infrasound became lower and lower.

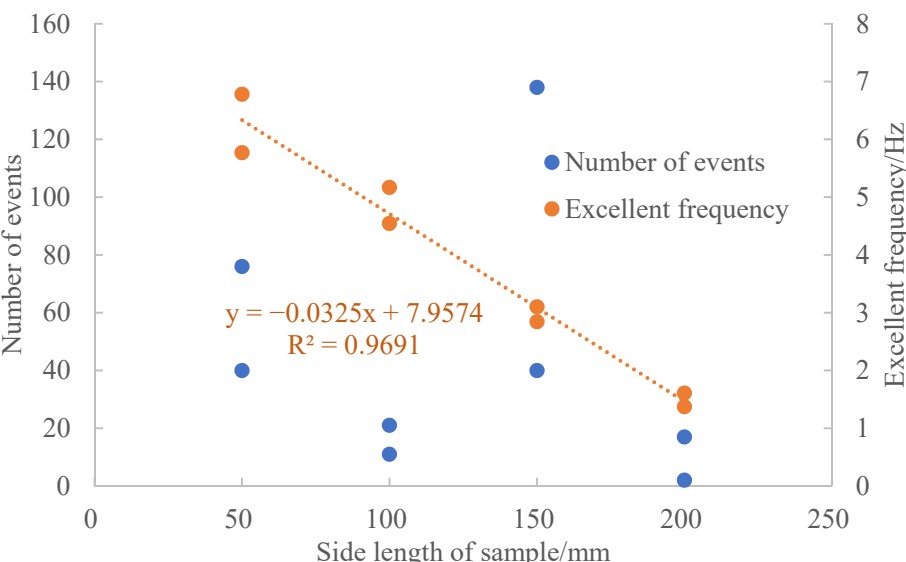

**Figure 9.** Number of infrasound events and average excellent frequency of samples with different sizes.

### 4.2. Impact of Sandstone Water Saturation on Infrasound Features in Shear Destruction

From the perspective of mechanics, during shear destruction, with increased water saturation, the shear resistance strength of the sandstone samples show a downward trend generally (as shown in Figure 10). This is because the authigenic clay mineral is an important interstitial material in sandstone and an important indicator mineral for the division of sandstone diagenesis stages. Clay minerals in rock absorb water and swell up, bringing about the concentration of force at the tips of microfissures, the structure damage between clay minerals, and a decrease in cohesion. The interference of water molecules also changes the physical state of rock, weakening the contact between particle surfaces. The results of experiments in this study are somewhat different from the traditional patterns [33]. The main reason is that the sandstone samples selected in this experiment were relatively tight and did not change much in saturation before and after being soaked in water.

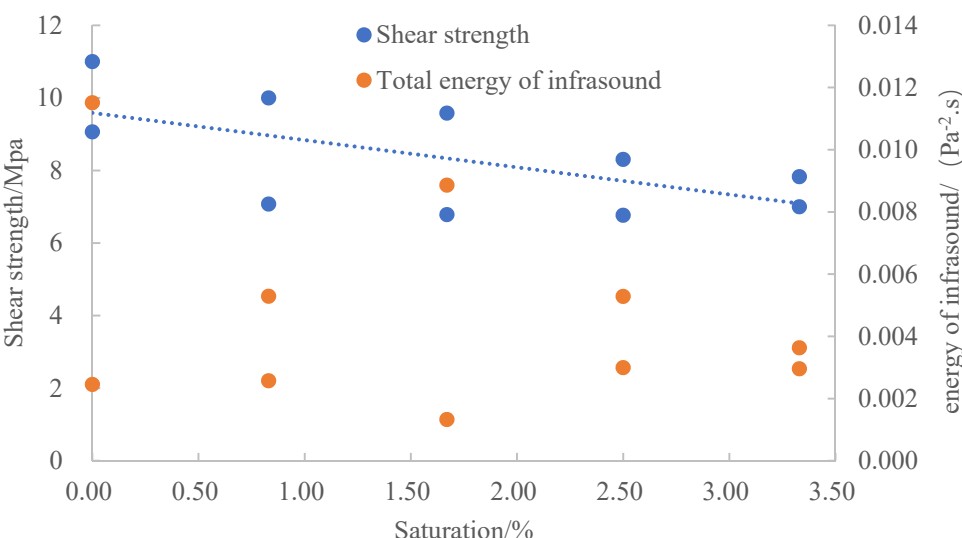

**Figure 10.** Relationship of shear resistance strength and infrasound energy with water saturation.

From the perspective of acoustics, with increased water saturation, the total energy (Figure 10) and the number of infrasound events (Figure 11) generated during the rock shear failure show little regularity. Since the rock samples in this study were tight sandstone ones and did not change much in water saturation before and after soaking, the total energy and number of infrasound events showed little regularity in variation. However, the average excellent frequency of infrasound events decreased linearly in general with increased water saturation. Affected by water saturation, the shear resistance strength of the rock decreased, and the shear failure of the rock became easier; therefore, the failure speed increased, the failure surface became flatter and flatter, and the corresponding excellent frequency of infrasound became smaller and smaller.

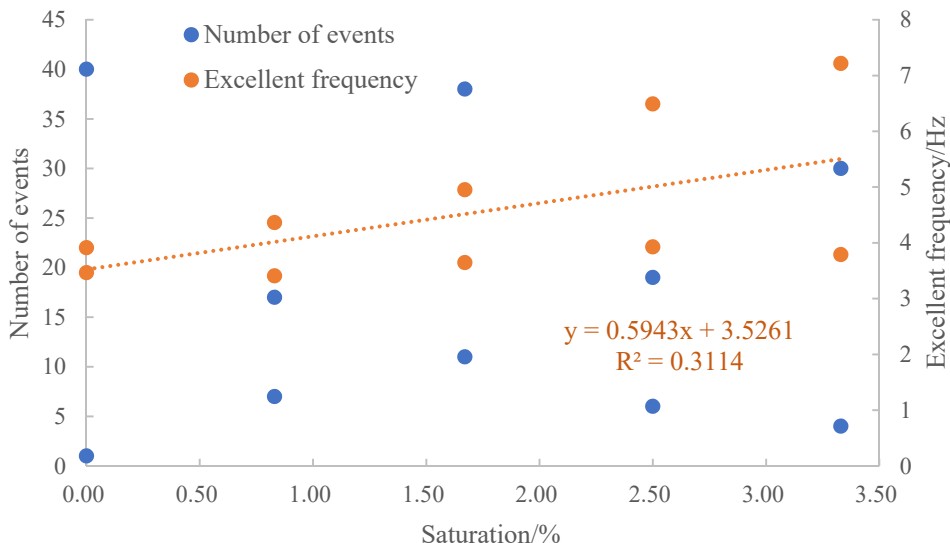

**Figure 11.** Relationship of number and average excellent frequency of infrasound events with water saturation.

### 4.3. Effect of Central Axial Pressure on Infrasound Features in Shear Destruction

From the perspective of mechanics, with increased axial pressure, the shear force on the rock increases linearly (Figure 12). The hardness of sandstone is related to its

mineral composition, structure and weathering degree. According to engineering rock mass classification standards (GB50218), the hardness coefficients of sandstone are F4, F8, F10 and F15, wherein the hardness increases from F4 to F15. As the sandstone is hard, its internal friction angle and cohesion hardly change, and its failure conforms to the Mohr–Coulomb strength curve.

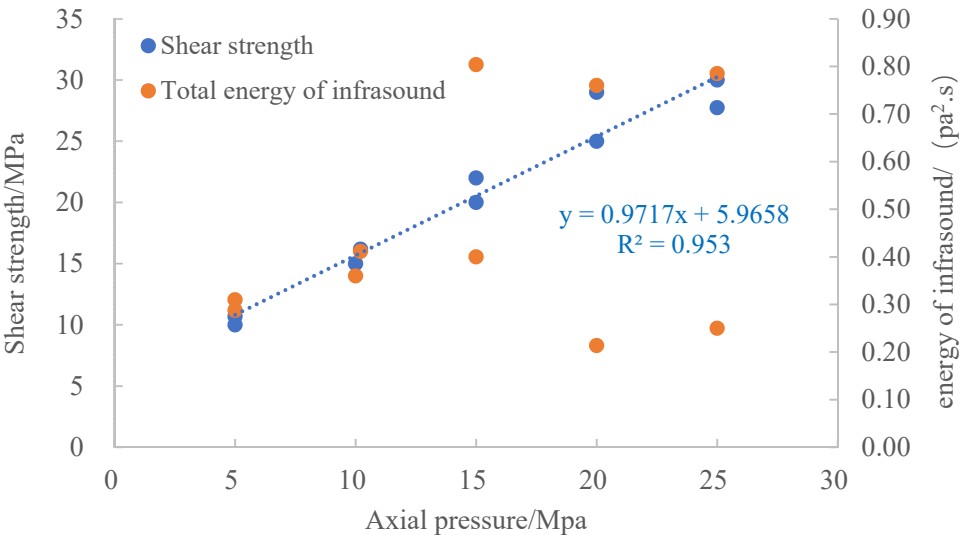

**Figure 12.** Relationship of shear resistance strength and infrasound energy with axial pressure.

We can see from the experimental results (Figure 13) that with increased axial pressure, the number of infrasound events increases linearly, and the average excellent frequency of infrasound events increases exponentially. When the axial pressure is small, the failure process is simpler, the process of generation, propagation, and running-through of micro-cracks in rock is short, the shear surface is relatively flat and straight (Figure 14a), the slip distance at the instant of failure is longer, so the corresponding infrasound is lower in frequency, and infrasound events are fewer. With increased axial pressure, the caving generated during the shear process increases in range and number (Figure 14b–e), and the infrasound events caused by caving also increases in number; the affected axial pressure, the broken surface of the sample is serrated, and the shear surface become increasingly rough, making the running through the slip distance shorter, so the corresponding infrasound generated is higher in frequency. With increased axial pressure, the energy released by rock failure increases, but the experimental results show the total energy of infrasonic signals generated by rock failure did not increase accordingly and had no regularity (Figure 12). This is probably because the samples in the experiments were small (all with a side length of 50 mm). Furthermore, the elastic wave generated during rock failure is higher in frequency, even exceeding the range of the infrasound frequency band. Hence, it is recommended to use sound emitting and infrasound means jointly in future studies to obtain a deeper understanding of the effect of axial pressure changes on sound wave energy in the process of shear loading on rock.

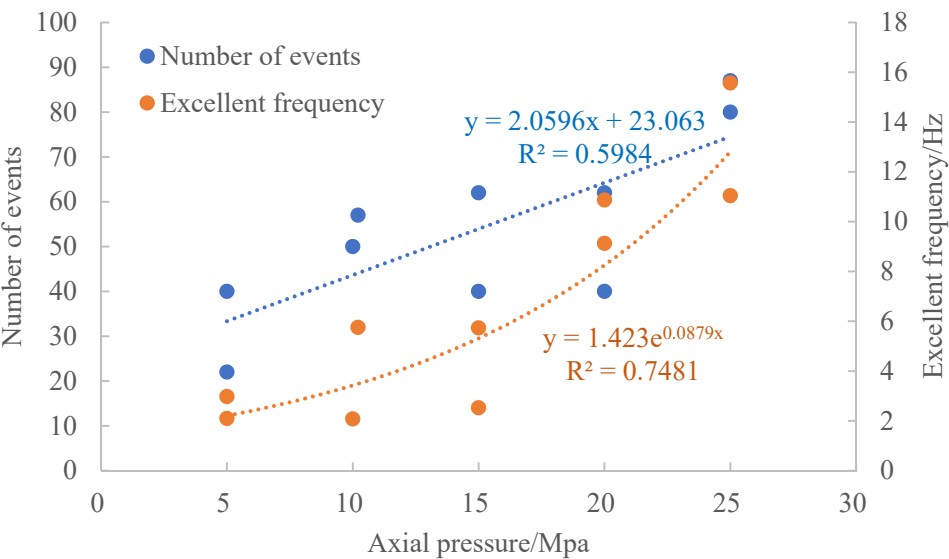

**Figure 13.** Relationship of the number and average excellent frequency of infrasound events with axial pressure.

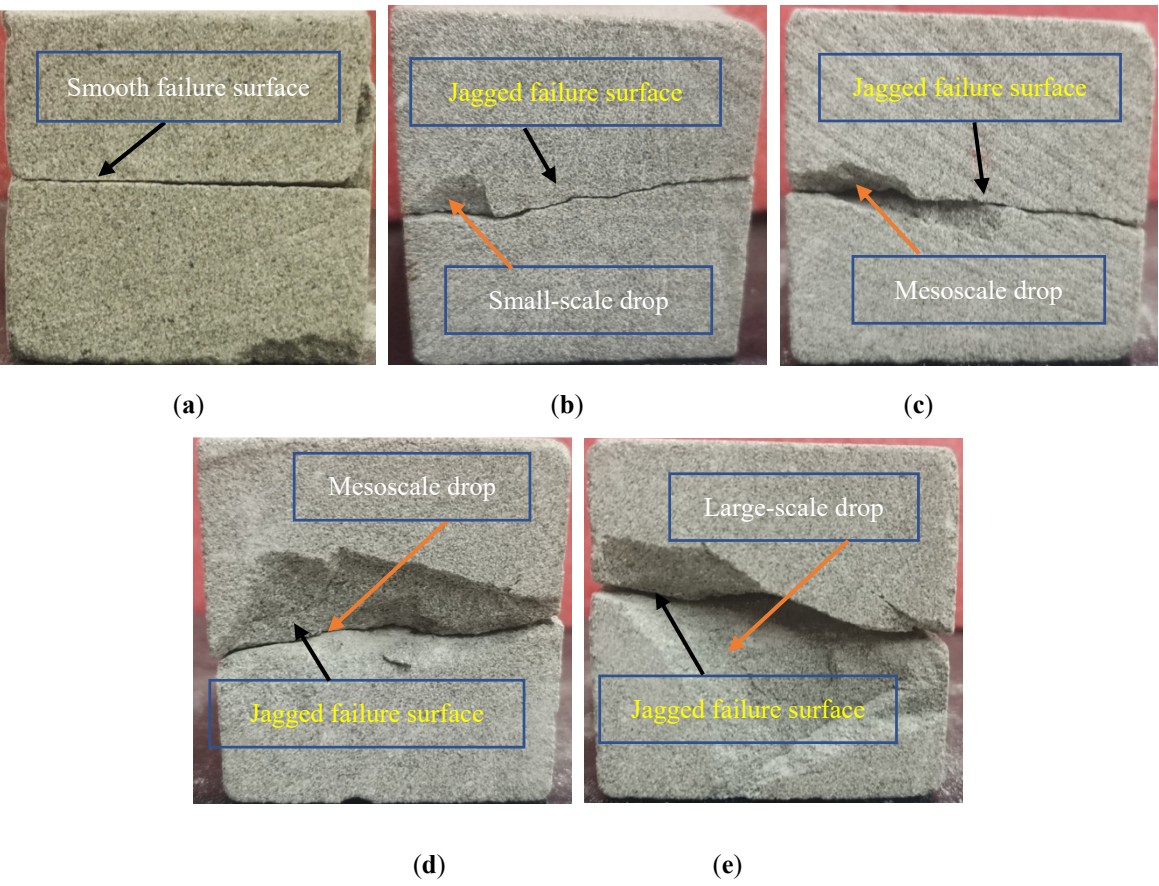

**Figure 14.** Direct shear experiment results of samples under different axial pressures. (**a**) Sample c1; (**b**) sample c2; (**c**) sample c3; (**d**) sample c4; and (**e**) sample c5.

## 5. Conclusions

To break the bottleneck in the monitoring and early warning of rocky landslides by infrasound, an experimental system collecting infrasonic signals generated during

sandstone shear has been built. Characteristic parameters of infrasonic waves generated by sandstone under shearing load were collected by using time-frequency analysis and automatic infrasound event pickup technology. The results show that: (1) the shear deformation of sandstone samples is always associated with infrasound events. The normalized energy cumulative coefficient changes violently, mainly at the beginning of elastic deformation and peak destruction of the sample. When the sample enters the elastic deformation stage, the normalized energy cumulative coefficient of the infrasound rises sharply for the first time with a steady increase. When the normalized energy cumulative coefficient is between 0.6 and 0.75, and the normalized infrasound ratio is between 0.89 and 0.93, the sample is just before the peak destruction, and this feature can be taken as the mark of early warning of rock failure; (2) the shear destruction of the sandstone sample is closely related to its size. With increased sample size, the shear resistance strength of sample increases logarithmically, the total energy of infrasound events increases exponentially, and the average excellent frequency of infrasound events decreases linearly. When the rock sample is small, destruction mainly occurs through slip failure. With increased rock size, the fractures of loading damage increase in length, and the corresponding excellent frequency of infrasound becomes smaller and smaller; (3) sandstone samples are quite tight, and do not change much in saturation before and after soaking; therefore, the total energy and number of infrasound events do not change much, but the average excellent frequency of the infrasound events goes down linearly with increased saturation in general. The modeling of the overburden pressure on the rock by axial pressure shows that with increased axial pressure, the number of infrasound events increases linearly and the average excellent frequency of infrasound events increases exponentially, but the total energy of infrasonic signals does not increase correspondingly.

According to the change law of infrasound characteristic parameters in the process of rock shear load, a method for judging rock stability by using infrasound parameters is established. According to this research idea, it is necessary to further carry out infrasound synchronous monitoring research of landslide physical model tests, which can provide a theoretical basis for the use of infrasound to carry out landslide warnings.

**Author Contributions:** C.Q., Writing—original draft; F.X., Y.L., Y.Z., H.H., J.C. and D.L. Data curation; P.S. Investigation; H.Z., Writing—review & editing; M.H., Funding acquisition. All authors have read and agreed to the published version of the manuscript.

**Funding:** This research received no external funding.

**Institutional Review Board Statement:** Not applicable.

**Informed Consent Statement:** Not applicable.

**Data Availability Statement:** The data used to support the findings of this study are available from the corresponding author upon request.

**Acknowledgments:** This research was supported by the Key Laboratory of Mine Geological Hazard Mechanism and Prevention and Control (no. 2018-08), Xigaze City Science and Technology Plan Project (RKZ2020KJ01), Key Laboratory of Rock Mechanics and Geohazards of Zhejiang Province (no. ZJRMG-2019-01), Chongqing Natural Science Foundation project (no. cstc2019jcyj-msxm0749, cstc2021jcyj-msxmX0187), Ecological environment survey and ecological restoration technology demonstration project in the water-level-fluctuating of the Three Gorges Reservoir (no. 5000002021BF40001), and The Tibet Autonomous Region Natural Resources Department's prevention and control capacity system construction project (no. 2020-0890-2). Their support is gratefully acknowledged.

**Conflicts of Interest:** The authors declare no conflict of interest.

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
