# Peer review of "Experimental Study on the Features of Infrasonic Waves of Sandstone under Shear Load"

_applsci, doi:10.3390/app112311552_

Round 1

Reviewer 1 Report

Missing of reference 23 in line 56.

Improving the description of water saturation process.

Improving the explanation of Figure 3. 

Show photos of samples A1-1 and A1-2.

Improving the quality of figures 4, 5 and 6 and put labels to explain.

In figure 6 subdivide the curves of graphics in different stages using letters for better explanation in the text.

The trend show in Figure 7 for total energy of infrasound is difficult to understand with the data obtained in the tests. For 200 mm samples there are two results and the higher are five or six times greater than the low, which one is correct?

The concept of “water on the particle surface acts as a lubricant, decreasing the friction” should be better explain.

The right scale of figure 10 must be revised.

Values of R2 of 0.3 (figure 10) of 0.6 (Figure 0.6)  not mean much, and some judgement must be done about such correlations.

The influence of water saturation in strength is well known and should be better explain in these tests as the results are weak.

The soundness of the statements (conclusions) much be improved taking into account the great variability of the results.

Reviewer 2 Report

Peer review report 1 On “Experimental study on the features of infrasonic wave of sandstone under shear load”

1. Original Submission

1.1. Recommendation

Major Revision

2. Comments to Authors:

Manuscript ID: applsci-1445606

Title: Experimental study on the features of infrasonic wave of sandstone under shear load

Authors: Qiao Chen, Xu Fenglin, Su Pengcheng, Liu Yang, Zhang Yifang, Zhu Honglin, and Huang Man

Overview and general recommendation:

The authors are trying to implement an interesting experiment for the early warning of slope failures that happened on sandstone rocks slopes.

A first comment reading carefully the paper is that the paper does fit the scope of the journal (publishment of experiments and their theoretical results in as much detail as possible), and it could be of interest to the readership of the journal. In addition, the experiments are described with sufficient details so as to allow another researcher to reproduce the results. However, there are some crucial points that are missing or are not written appropriately in the whole paper.

Therefore, I recommend major revision. I explain my points in more detail below. Thus, to make this paper publishable, the authors need to respond to the following remarks.

2.1. Major comments:

  1. Authors base their experiments on rocks which are consisted of sandstone types. My suggestion to authors is prior to the description of their experiments to write a paragraph (or more than one) in which they will name some examples with sandstone slopes that result in landslides. To my knowledge, it is more usual for a geologist/geotechnical engineer to come up with slope failures manifested due to the appearance of a combination of sandstone along with siltstone or limestone/chert rocks rather than sandstone ones alone. However, if authors indeed have found slope failures that have happened only by stability loss of sandstone slopes, they should refer to some examples (including references) from China or elsewhere in the world.

  2. Writing some examples of already sandstone slope failures case studies, authors will explain better their scope of trying to investigate landslide early warning from a different perspective by executing the described experiments they report in their work. This suggestion of mine could be added at the final part of their introduction.

  3. Page 3, Line 113: Please explain more analytically what do you mean by this phrase (e.g. “To determine the effects of different overlying stratum pressure on sandstone slope, …). For instance, do you mean that some other petrology type of rocks overly, somehow, sandstone rocks which are the basic rock of the examined slopes? In addition, give an example.

  4. Page 6, Line 171: Reading unit 3.2 (Automatic collection of infrasound events), I wonder if there are any references when authors describe the mentioned steps 1 and 2.

  5. Page 8, Lines 257-260: Please give an example and some references.

  6. Lines 261-262: Please give some references.

  7. Page 9. Line 280: Please explain more analytically, how these conclusions lead to the judgment of the stability of the sandstone slopes (e.g. “These conclusions can help judge the stability of the sandstone slope”). For example, have these conclusions come out by comparing the paper’s results to other laboratory tests from different case studies of the world respectively?

  8. Page 12, Line 318: To my knowledge, the majority of the mineral composition of sandstones is made of such ingredients which have nothing to do with clay minerals. Thus, I can not understand the phrase of line 318 (e.g. “This is because clay minerals in rock absorb water and swell up, …..”), regarding its (e.g. clay) relationship with the appearance of sandstone rock in a slope.

  9. Line 325: What do authors mean by the word “tight”? Do they mean impermeable?

  10. Line 340: Please define which type of “hard” sandstone this research is focused on because there are different types of sandstones (from the point of view of petrology/mineralogy as well as geotechnically).

  11. Line 396-397: Please explain better the phrase “The modeling of overburden pressure on rock by axial pressure shows that with the increase of axial pressure, …).

  12. Regarding the last unit (5. Conclusions), my suggestion is that an additional statement should be written on how the above-mentioned experiment methodology could be taken into consideration by the geological/geotechnical scientific community for landslide early warning monitoring.

22. Minor comments:

  1. Line 56: Please add the number of references [23]. Also, the reference with the number 24 is missing in the passage.

  2. There is poor resolution on the following figures (e.g. Fig. 1, Fig. 4, Fig. 5, Fig. 6.). In addition, in Fig. 5 (regarding what it is said in line 164) red color is not clear.

  3. Line 178: Please replace “humen” with “humans”

  4. Line 235: Replace 3.4 with 3.3.

  5. Line 328: Where can somebody in Fig. 10 or Fig. 11 realize that infrasound events decreased linearly?

  6. Lines 413-467: Authors should recheck the proposed reference format of the journal.

In conclusion, the novelty of the article and its impact on the field of landslides is medium and the revisions are too fundamental for the submission to continue being considered in its current form. The paper needs changes such as:

(a) the analysis in some parts must be presented in a more appropriate way in order for authors to explain better the presented hypotheses and speculations,

(b) referring to more references (as pointed out above),

(c) some above-mentioned figures must be presented more clearly, and finally

(d) the conclusions should be more interesting for the readership of the Journal of Applied Sciences, not just mentioning by bullets the most interesting findings from the experiments, but also the Conclusions should start by an introductory paragraph and end up with a final argument that summarizes what the authors refer to the last two sentences of their abstract.

Round 2

Reviewer 1 Report

Taken into account the answers provided

(1) Reference 23 ok. There is no references in line 42 and 181.

(2) The laboratory procedure used in rock saturation is not provided.

(3) ok

(4) ok

(5) No labels?

(6) Identified the elastic, plastic and destruction stages in the figures. Explain the meaning of the black dashed line in the figure 6.

(7) So, what is it the influence of sample size as other factors are more important and are superimposed. What conclusions may be considered?

This topic is not well explain.

(8) The mineralogical composition of sandstone studied is not provided, but as a sandstone most of the grains must be quartz, which is very stable, and so "the destruction of rock grains" should be negligible. The explanation provided is insufficient and should be rewrite or remove.

(9) ok

(10) The sentence between lines 308 and 313 should be rewritten or removed. The role of saturation in the experiments is not well explain. It will better only to stat the facts than to try explain the reasons with not convinced arguments.

(11) Already comment previously.

(12) The conclusions should be rewritten and some terminology avoid as for example "tight" as the meaning is not clear. It may be stressed that based on the data presented the soundness of the statements is not strong and some caution must be taken in the draw of the conclusions.

As a overall comment some explanations of the phenomena are weak and either they must be improved or the paper will be better without those ones.

Reviewer 2 Report

Peer review revised report On “Experimental study on the features of infrasonic wave of sandstone under shear load”

  1. Original Submission

1.1. Recommendation

Minor Revision

  1. Comments to Authors:

Manuscript ID: applsci-1445606

Title: Experimental study on the features of infrasonic wave of sandstone under shear load

Authors: Qiao Chen, Xu Fenglin, Su Pengcheng, Liu Yang, Zhang Yifang, Zhu Honglin, and Huang Man

Overview and general recommendation:

Auditing carefully authors’ revised answers and in responding to Reviewer 2 comments, it can be clearly expressed that authors have integrated thoroughly their revised answers into the whole manuscript by providing (to me) satisfactory explanations.

However, concerning the revised manuscript, there are some additional remarks - points that authors should amend before this paper regarded as publishable.

Minor comments:

  1. Lines 78-85: For better understanding of this paragraph, please insert a map with the area you describe in lines 78- In addition, regarding the phrase “These methods have a smaller monitoring range, while infrasound monitoring has a wider range”, please give a reference.
  2. Lines 250-252: How the mentioned values (0.6, 0.75, 0.89, 0.93) of these two acoustic parameters can be correlated with the decision of whether the sandstone is in the near failure stage? Are there examples from literature or are there characteristic (e.g. standards) laboratory threshold values that can be compared with the authors’ findings, in order to determine the above-mentioned?
  3. Lines 309-311: Please add the answer you gave in your cover letter (“The authigenic clay mineral is an important interstitial material in sandstone and an important indicator mineral for the division of sandstone diagenesis stages”), in those two lines (309-311).
  4. Line 339: Please add the answer you gave in your cover letter (“The hardness of sandstone is related to its mineral composition, structure and weathering degree. The hardness coefficients of sandstone are F4, F8, F10 and F15, the hardness is getting higher and higher from F4 to F15”), in lines 338- In addition, define the symbology you refer to (e.g. F4, F8, F10, F10 & F15) according to some references.
  5. Conclusions (Section 5), should be rewritten based on my last comment of my previous review (e.g. the conclusions should be more interesting for the readership of the Journal of Applied Sciences, not just mentioning by bullets the most interesting findings from the experiments, but also the Conclusions section should start by an introductory paragraph and end up with a final argument that summarizes what the authors refer to the last two sentences of their abstract).

In conclusion, I have no further comments as I am totally satisfied with their answers and their modifications they made in the revised manuscript.
